# Genetic Diversity, Ochratoxin A and Fumonisin Profiles of Strains of *Aspergillus* Section *Nigri* Isolated from Dried Vine Fruits

**DOI:** 10.3390/toxins12090592

**Published:** 2020-09-14

**Authors:** Petra Mikušová, Miroslav Caboň, Andrea Melichárková, Martin Urík, Alberto Ritieni, Marek Slovák

**Affiliations:** 1Plant Science and Biodiversity Centre, Institute of Botany, Slovak Academy of Sciences, Dúbravská cesta 9, SK-845 23 Bratislava, Slovakia; miroslav.cabon@savba.sk (M.C.); andrea.melicharkova@savba.sk (A.M.); marek.slovak@savba.sk (M.S.); 2Institute of Laboratory Research on Geomaterials, Faculty of Natural Sciences, Comenius University in Bratislava, Ilkovičova 6, SK-842 15 Bratislava 4, Slovakia; martin.urik@uniba.sk; 3Department of Pharmacy, School of Medicine, University of Naples Federico II, Staff of Unesco Chair for Health Education and Sustainable Development, 801 31 Napoli, Italy; alberto.ritieni@unina.it; 4Department of Botany, Charles University, Benátská 2, CZ-128 01 Praha 2, Czech Republic

**Keywords:** food spoilage, mycotoxin, fungal diversity, calmodulin, beta-tubulin, HPLC

## Abstract

We investigated ochratoxin A (OTA) contamination in raisin samples purchased from Slovak markets and determined the diversity of black-spored aspergilli as potential OTA and fumonisin (FB1 and FB2) producers. The taxonomic identification was performed using sequences of the nuclear ITS1-5.8s-ITS2 region, the calmodulin and beta-tubulin genes. We obtained 239 isolates from eight fungal genera, of which 197 belonged to *Aspergillus* (82%) and 42 strains (18%) to other fungal genera. OTA contamination was evidenced in 75% of the samples and its level ranged from 0.8 to 10.6 µg/kg. The combination of all three markers used enabled unambiguous identification of *A. carbonarius*, *A. luchuensis*, *A. niger*, *A. tubingensis* and *A. welwitschiae*. The dominant coloniser, simultaneously having the highest within-species diversity isolated from our raisin samples, was *A. tubingensis*. Out of all analysed strains, only *A. carbonarius* was found to produce OTA, but in relatively high quantity (2477–4382 µg/kg). The production of FB1 and FB2 was evidenced in *A. niger* strains only.

## 1. Introduction

Dried vine fruits are some of the most favourite and frequently utilised ingredients in the food industry; however, they are often exposed to contamination by metabolites of various microorganisms, including filamentous fungi [1]. The fungal genus *Aspergillus* comprises numerous species which significantly impact overall food production and human health [2]. Out of them, members of the *Aspergillus* section *Nigri* [3] are considered to be some of the most influential food contaminants of dried vine fruit products worldwide. These microorganisms are well known as important producers of a large scale of mycotoxins, but especially ochratoxins and fumonisins [4]. Indeed, members of the *Aspergillus* section *Nigri* were proven to be the main OTA (ochratoxin A) producers in grapes and are responsible for the contamination of wine, grape juice and raisins [5,6,7,8]. OTA is one of the most harmful and most strictly monitored mycotoxins [9,10]. The European Commission determined the maximum OTA limit for raisins at 10 µg/kg [11]. The accurate number of black aspergilli species producing OTA is still enigmatic because of uncertainty in their identification [2,12]. Likewise, fumonisins also belong to toxic secondary metabolites produced by members of black aspergilli species and can be considered a possible source of mycotoxin contamination. The classification of fumonisins is based on their structure, resulting in four classification groups—namely A, B, C and P. The most frequent and abundant in food commodities are, however, only FB1, FB2 and FB3 [13,14]. Importantly, FB1 and FB2 were found to be potentially carcinogenetic, and their presence in food was associated with a high incidence of human oesophageal cancer in China and South Africa [15,16]. The fumonisins were evidenced to be produced predominantly by specific members of the genus *Fusarium* (e.g., [4,13]). Nevertheless, their production has also been evidenced in some members of the *Aspergillus* species, specifically in *A. niger,* and rarely also in *A. welwitschiae,* to date [17,18,19,20].

Correct determination of the members of *Aspergillus* section *Nigri* has also been challenging for specialists in the taxonomy and evolution of the genus *Aspergillus* [21,22]. The taxonomy of this species group has been intensively studied by many taxonomists, and the most current taxonomic concept of black aspergilli recognises 27 well-delimited species [2,23,24,25]. The pivotal problem hampering unambiguous delimitation of black aspergilli is embodied in their cryptic speciation. Due to the lack of taxonomically diagnostic features, black aspergilli are hardly distinguishable by traditional micro- and macromorphological approaches [2,26,27]. Various DNA fingerprinting methods [28,29] and/or utilisation of low-copy protein-coding genes were shown to be the most efficient tools for reliable identification of species from the *Aspergillus* section *Nigri* [2,8,27].

In the present study, we investigated OTA contamination of dried vine fruit packages of various origins purchased in Slovak markets and analysed the taxonomic diversity and the ochratoxin and fumonisin (FB1 and FB2) production ability of the black aspergilli strains isolated. The following questions were addressed:(1)What is the level of OTA in the analysed dried vine fruit samples, and does the OTA content exceed the critical maximum limit of 10 µg/kg?(2)What is the overall diversity of microscopic fungi colonising the dried vine fruit samples? How many species from the *Aspergillus* section *Nigri* can be isolated and identified in the tested samples?(3)What are the individual ochratoxigenic abilities of selected isolated strains of members of *Aspergillus* section *Nigri*? Do some isolated black aspergilli strains produce fumonisins FB1 and FB2?

## 2. Results

### 2.1. Cultivations and Morphological Identifications

The initial screening of the analysed dried vine fruit samples revealed that, except two samples, all were colonised by microfungi. The level of contamination ranged between 2.7 × 10^2^ and 3.6 × 10^3^ CFU/g. The direct plating method onto malt extract agar (MEA) agar recovered 239 isolates, representing 14 taxa and belonging to 8 genera of *Alternaria*, *Cladosporium*, *Aspergillus*, *Paecilomyces*, *Penicillium*, *Rhizopus*, *Saccharomyces* and *Trichothecium* (Table 1). Based on the macro- and micromorphological identifications, only the strains of *Alternaria alternata*, *Cladosporium cladosporioides*, *Penicillium chrysogenum* and *Trichothecium roseum* were identified unambiguously at the species level.

The predominating colonisers, found in 90% of the samples, belonged to the genus *Aspergillus*. Out of the 239 isolated strains, 197 represented the genus *Aspergillus* (82%), while only 42 strains (18%) were assigned to other fungal genera. The percentage of aspergilli in the CFUs (colony forming units) oscillated was between 0 and 100%. The initial morphological evaluation indicated that our strains belonged to *Aspergillus* section *Flavi* (*A. flavus*) and section *Nigri* (several species, Table 1).

### 2.2. Molecular Analyses of the Black Aspergilli Strains

After concatenation of all markers, we recovered 38 sequences, but in five cases, at least one marker was missing (G85—no beta-tubulin (benA hereafter) and ITS1-5.8s-ITS2 (ITS hereafter); G131—no ITS and calmodulin (CaM hereafter); G189—no CaM; G203—no ITS and G212—no benA). The direct comparison of edited sequences with trimmed ends, but with indels and ambiguously aligned regions, revealed: (1) four ribotypes in the ITS region (36 sequences); (2) twelve haplotypes in the CaM gene (36 sequences); (3) twenty-four haplotypes in the benA gene (35 sequences) (Appendix A). All three gene datasets included strains assignable only to biseriate species (groups) of black aspergilli (Appendix A).

Altogether, we identified twenty-six genotypes that were assignable to five biseriate species (groups) of black aspergilli (Figure 1; Appendix A). Two strains with two genotypes belonged to *Aspergillus carbonarius*, six strains with five genotypes were identified as *A. luchuensis*, four strains with two genotypes belonged to *A. niger*, nineteen strains with fourteen genotypes were assigned to *A. tubingensis* and, finally, three strains with three genotypes belong to *A. welwitschiae*. The comparison of our strains with those deposited in the National Centre for Biotechnology Information (NCBI) was unambiguous since the identity values reached more than 98% (at least in benA and CaM).

The maximum likelihood analysis based on the concatenated data matrix resulted in a phylogenetic tree composed of three well-supported major clades (Figure 1). The first clade composed of an outgroup *Aspergillus flavus* only, while the second one included all uniseriate species. The last and most complex third clade encompassed the rest of the black aspergilli together with all of our samples. Our strains appeared, thus, in the biseriate clade and can be divided into two distinct genetic groupings. Two of our strains (two genotypes) were clustered in a single subclade together with *A. carbonarius* and, genetically more distinct, *A. ibericus*, *A. sclerotiicarbonarius* and *A. sclerotioniger*. The rest of our strains appeared in several small clusters within the largest terminal subclade. Four strains (two genotypes) were identical with *A. niger.* Three other strains (three genotypes) were related to the *A. welwitschiae* type strain, with one clustered directly with *A. welwitschiae* and two that formed small subclades in sister positions. Nineteen strains (fourteen genotypes) were co-clustered with *A. tubingensis*; and finally, six strains (five genotypes) were placed together with the type strains of *A. luchuensis.*

Although five strains lacked one or two markers, all of our samples could be unambiguously assigned to a particular species (Figure 1; Appendix A).

### 2.3. Determination of Ochratoxin A (OTA) Content in Analysed Raisins Samples

Ochratoxin A was detected in 15 samples (75%) of analysed dried vine fruits (Table 1). Only five samples (originating from the Czech Republic, Chile, Iran, Slovakia, and Turkey) were not contaminated. The amount of OTA in the positive samples varied between 0.8 and 10.6 µg/kg. In two samples originating from Chile, the OTA level exceeded the critical maximum limit of 10 µg/kg prescribed in the European Union (EU) regulations. Indeed, OTA contamination was found in all world regions from which our raisin samples originated.

### 2.4. Toxigenic Ability of Aspergillus Isolates—Production of Ochratoxin A and Fumonisins B1 and B2

Out of all analysed isolates of *Aspergillus* section *Nigri*, only two isolated strains of *A. carbonarius* (G_187, G_191) were found to be ochratoxigenic (Appendix A). OTA production reached relatively high quantities of 2477 and 4381 µg/kg, respectively. The ability to produce FB1 and FB2 was evidenced only in *A. niger* and, again, in all its isolated strains (G_033, G_050, G_209, G_210; Appendix A). FB1 production ranged between 510 and 1213 µg/kg, whereas the amounts of FB2 were remarkably higher, ranging between 5509 and 11,118 µg/kg (for details see Appendix A). The strain G_033 produced pronouncedly higher amounts of both fumonisins compared to the other three strains.

## 3. Discussion

### 3.1. Taxonomic Identity of Isolated Strains and Diversity of Black Aspergilli on Dried Vine Fruits

The micromycetes colonising the dried vine fruit samples analysed in this study belonged to seven genera of microscopic filamentous fungi, specifically *Alternaria*, *Cladosporium*, *Aspergillus*, *Paecilomyces*, *Penicillium*, *Rhizopus* and *Trichothecium*. In addition, we also isolated yeasts from the genus *Saccharomyces*. Nevertheless, the vast majority of strains belonged to members of the *Aspergillus* section *Nigri*. The combination of three genetic markers used allowed us to identify and unambiguously assign all of our black aspergilli strains to five species from the biseriate group (*A. carbonarius*, *A. luchuensis*, *A. niger*, *A. tubingensis* and *A. welwitschiae*). In general, the delimitation of members of black aspergilli is not straightforward due to the lack of phenotypic diagnostic characters, which is a direct consequence of their fast diversification and speciation [2,21,30]. This is especially true in the case of the taxonomically difficult, cryptic species from the *A. niger* group, of which unambiguous identification is possible only using fingerprinting methods and/or a combination of protein-coding genes [2,27,28,29,31].

Fourteen out of the twenty-seven currently recognised black aspergilli have been documented as colonisers of dried vine fruits up to date [5,32,33,34,35]. Nevertheless, only a few studies reported more than five species from one type of substrate [31,32,33,34,35,36]. The most dominant and, simultaneously, most genetically diverse species in our study (fourteen genotypes) was *A. tubingensis*. This finding corresponds with the outcomes of recent investigations identifying *A. tubingensis* to be the predominant coloniser of dried vine fruits [7,28,31]. Older publications indicated the prevalence of *A. niger* agg. members on raisins, but this is not in conflict with previous statements since *A. tubingensis* belongs to this species group and might remain undetected due to the lower resolution of genetic markers used [21,37,38]. The real number of black-spored aspergilli colonising dried vine fruits might be, in general, underestimated, and this problematic calls for further comprehensive studies involving a combination of methodological approaches, including multigene analyses.

### 3.2. OTA Contamination of Analysed Dried Vine Fruits Samples

Ochratoxin A is polyketide mycotoxin considered to be one of the most dangerous secondary metabolites produced by various species of the genera *Aspergillus* and *Penicillium* [6,39,40,41]. Since its discovery in 1965 [42], contamination with OTA has been detected in a broad range of food commodities, such as coffee, legumes, cereals, meat and fresh and dried fruits [26,39,40,43,44,45,46,47]. The majority of notifications reported in the RASFF (Rapid Alert System for Food and Feed) concerned aflatoxin B1 (80%), followed by aflatoxins (not specified) (13%) and OTA (5%). The notification trend of OTA during 2007–2016 shows a significant increase, with a peak of notifications in 2016 (data from portal of RASFF) [48]. We found traces of OTA contamination in 75% of the raisin samples (Table 1). Such a high level of contamination is, however, not unexpected in this type of food commodity. Results of previous studies showed that OTA contamination in analysed dried vine fruit samples ranged between 20 and 100% (e.g., [49,50,51]). We are, however, slightly reluctant to broadly generalise our findings, as the precise percentage of OTA-contaminated samples in our study might be affected by the somewhat restricted number of analysed raisin samples. The levels of OTA in our dried vine fruit samples were relatively low, ranging between 0.8–10.6 µg/kg. The maximum tolerable value of 10 µg/kg established by the European Commission was, thus, exceeded only in two samples from Chile (sample no. 9 with 10.5 µg/kg and no. 13 with 10.6 µg/kg; Table 1). This finding corresponds well with results of numerous investigations which indicated that, despite the high incidence of OTA in the analysed samples, only a low percentage of the contaminated samples exceeded the maximum tolerable values [50,52,53,54,55,56]. Nonetheless, high levels of OTA in dried vine fruits, exceeding the critical threshold value of 10 μg/kg, were documented elsewhere [31,43,51,57,58,59]. In fact, the detected OTA contamination levels in raisins significantly varies among particular case studies. Such a variation in OTA content used to be predominantly attributed to changes in environmental and climatic conditions [53]. The influences of warm weather, latitude, presence of high humidity and rainfall, especially before the harvest time, were considered as factors responsible for the increased production of OTA in raisins [35,60,61].

### 3.3. Ochratoxigenic and Fumonisins Production Potential of Aspergillus Section Nigri Isolates

Members of *Aspergillus* section *Nigri* were proven to be predominant OTA producers and contaminants of vine fruits and associated products (e.g., [1,7,25,29,37,51,62,63]). The ability of our strains to produce OTA was apparently limited since only strains of *A. carbonarius* proved to be ochratoxigenic. Interestingly, there was no correlation between the presence of *A. carbonarius* in specific samples and their OTA contamination levels. Both analysed strains produced OTA in relatively high quantities, which is in line with the outcomes of previous studies (e.g., [7,33,37]). Indeed, *A. carbonarius* has been recurrently evidenced to be the most prominent source of OTA contamination in all types of vine fruits (e.g., [6,7,25,31,40,57,64,65]). The ochratoxigenic ability of *A. carbonarius* could be affected by various environmental factors, and its incidence varied significantly among particular case studies [6,23,58,66,67,68,69,70].

On the other hand, our study revealed that the remaining strains belonging to *A. luchuensis*, *A. niger*, *A. tubingensis* and *A. welwitschiae* did not produce OTA within in vitro tests. This finding is at least partially contradictive to several previous investigations detecting OTA production, at least in some members of the *A. niger* aggregate [7,37,57,71]. For instance, OTA production was detected in strains of *A. niger* and *A. welwitschiae* originally isolated from vine fruit products [1,7]. On the other hand, both species were shown to be only weak to moderate OTA producers, at least compared to *A. carbonarius*. The most abundant species isolated from our raisin samples was *A. tubingensis*, including both samples highly contaminated by OTA (Table 1). Although several authors previously declared the ochratoxigenic potential of *A. tubingensis* strains [24,28,72], the recent genomic analysis of this species revealed the absence of genes responsible for OTA production [73]. Thus, *A. tubingensis* cannot be a source of OTA contamination in our dried vine fruit samples. Likewise, *A. luchuensis* was also reported to be a species incapable of OTA production [62]. The detected presence of OTA in our dried vine fruit samples, thus, most probably originated from another OTA-producing micromycete which was left undetected.

Fumonisins belong to mycotoxins produced predominantly by species from the genus *Fusarium*, but especially by *F. verticillioides* and *F. proliferatum* [13,74]. Nonetheless, production of fumonisins was also repeatedly evidenced in a few members of *Aspergillus* section *Nigri* [17,18,19,20]. All of the four isolated strains belonging to *A. niger* were shown to be FB1 and FB2 producers. This finding is in good agreement with previous studies indicating that out of all species from *Aspergillus* section *Nigri,* the most potent FB1 and FB2 producer is *A. niger* [17,18,19,20,75]. Another possible candidate capable of producing fumonisins is *A.*
*welwitschiae* [17,18,19,20,65,72]. Nevertheless, we did not find traces of FB1 and FB2 production by our *A.*
*welwitschiae* strains. Chiotta et al. (2011) [72] investigated the incidence of black aspergilli in Argentinian grapes and identified their OTA and fumonisin production abilities. The most critical fumonisin producer (FB2–FB4) in their study was *A. niger*, however, more interestingly, the authors also isolated a strain capable of producing both OTA and fumonisins.

## 4. Conclusions

We provided evidence that the contamination of dried vine fruits by microscopic fungi and their mycotoxins is a global and long-term persisting problem in the food industry. Black spore aspergilli are well-known spoilers and mycotoxin contaminants of grapes and associated products. The vast majority of analysed raisin samples were colonised by members of *Aspergillus* section *Nigri*, which emphasises how severe a problem such contamination might be for ordinary consumers. On the other hand, the critical limits were exceeded only in two cases. A combination of three genetic markers enabled us not only to identify five black spore aspergilli species, but also to reveal remarkable intraspecific genetic diversity within the most frequently isolated *A. tubingensis* strains. Multimycotoxin profiling uncovered the ochratoxigenic ability of *A. carbonarius* strains, while fumonisin (FB1 and FB2) production was detected in *A. niger* isolates. Our investigation represented an essential contribution to the problem of fungal and mycotoxin contamination of raisins. It is crucial to continuously inspect the presence of possible fungal contaminants in this type of food commodity and to determine traces of their potentially toxic secondary metabolites.

## 5. Material and Methods

### 5.1. Sampling

Our sampling design included all dried vine fruits brands available in Slovak food markets during the period from March to August 2016 (Table 1). Altogether, we analysed 20 samples (packages) of dried vine fruits originating from several world production areas (continents): Asia (8 samples), Africa (1 sample), South America (6 samples), Europe (3 samples) and two samples of unknown origin, unpacked and directly bought at a fruit market (Table 1). These samples represented all types of sold dried vine fruits (sultanas, raisins and currants), and none of them reached their expiration date before cultivation and ELISA analysis. Before analyses, all samples were stored at 7 °C to prevent possible biodegradation.

### 5.2. Isolation and Morphological Identification of Retrieved Strains

For cultivation experiments, we selected 150 intact dried vine fruits per sample. The fruits were plated directly onto malt-extract agar (MEA), following the study [4]. The plates were incubated at 25 ± 1 °C for 14 days and inspected daily. After incubation, all plates were visually inspected, and all developed fungal colonies were morphologically identified under a Carl Zeiss Scope A1 Axio optical microscope. Colonies of black aspergilli were extracted from the plates and transferred onto MEA and Czapek–Dox agar (CZDA) for further identification. Pure cultures were retrieved using the standard dilution technique and sub-cultured on a suitable medium, as described by [4]. The quantity of microscopic fungi was detected in 1 g of dried vine fruits according to the degree of contamination, i.e., the colony-forming unit (CFU/g) and frequency (percentage) of the *Aspergillus* within the CFU (Table 1).

All obtained cultures of black aspergilli were morphologically inspected and clustered, according to their development and macro- and micromorphological characteristics, into morphotype strain groups. The genetic diversity of the black aspergilli from the dried raisin samples was determined on two strains per sample, taking into account the morphotype group. The strains were randomly selected and subjected to genetic and toxicological analyses. All pure retrieved cultures were diluted in 20% glycerol and stored at ‒70 °C as conidial suspensions and deposited at the Institute of Botany, Plant Science and Biodiversity Centre, Slovak Academy of Sciences.

### 5.3. Molecular Analysis

Before DNA extraction, a volume of 500 µl of conidial suspensions was taken from the selected strains and diluted in 50 mL of the Sabouraud + peptone liquid medium and incubated for 3 days at 25 °C with 150 rev/min shaking. Genomic DNA was isolated from the pre-prepared samples using the DNAeasy Plant Mini kit (Qiagen, Inc., Valencia, CA, USA) according to the manufacturer’s recommendations. The precise identification of the isolated strains was performed using a combination of three molecular markers. Specifically, we used the ITS region (ITS1-5.8S-ITS2) of the nuclear ribosomal DNA and two nuclear protein-coding genes, namely calmodulin (CaM) and beta-tubulin (benA) [2]. The amplification was performed using the following primer combinations: ITS region—ITS1 and ITS4 [76]; CaM—CMD5 and CMD6 [77]; benA—Bt2a and Bt2b [78]. All markers were amplified with PCR beads (Hot Start MIX RTG, GE Healthcare), using the thermal-cycling protocol, according to [2]. PCR products were enzymatically purified by employing the PCR Product Clean-Up Prior to Sequencing kit (Exonuclease I and FastAP thermosensitive alkaline phosphatase, Thermo Fisher Scientific). Purified PCR products were sequenced at GATC Biotech AG, European Custom Sequencing Centre, Cologne, Germany.

### 5.4. Sequence Alignments and Phylogenetic Analysis

Sequences were edited in Geneious v.10 (Biomatters, Auckland, New Zealand) and aligned using the MAFFT algorithm [79]. Occasional single nucleotide polymorphisms were labelled with NC-IUPAC ambiguity codes. The precise identification of exons and introns in the CaM and benA genes was made using the annotations of the *A. nidulans*_FGSC_A4 genome available in the aspergillus genome database (AspDG; http://www.aspergillusgenome.org/). Ambiguously aligned positions detected in the sequences of all tree genetic markers were removed using Gblocks ver. 0.91b [80]. Remaining indels were considered as missing data. Two data matrices, including newly generated sequences from our strains and selected sequences of all currently accepted members of the *Aspergillus* section *Nigri* retrieved from the NCBI, were prepared. ITS region, CaM and benA sequences of 27 species from the section *Nigri* and one sequence of *A. flavus* serving as an out-group were co-analysed (Appendix A). The exception represented *A. trinidadensis* and *A. floridensis*, for which only CaM and benA sequences were available in the GenBank. We decided to take advantage of the synergistic effect of combining datasets and analysed only concatenated alignments. The first data matrix was composed of concatenated ITS, CaM and benA sequences with trimmed ends, but with indels and ambiguously aligned regions included. The second data matrix encompassed the same dataset, but sequences were edited by Gblocks ver. 0.91b.

The genetic divergence and variation of our strains were analysed in two steps. Firstly, the genetic divergences of the sequences of strains and their tentative assignment to already described species were based on the edited ITS, CaM and benA sequences with trimmed ends, but including indels and ambiguously aligned regions. The sequences were directly compared with those from the *Aspergillus* section *Nigri* deposited in the NCBI GenBank database using the BLAST search tool (https://blast.ncbi.nlm.nih.gov/Blast.cgi). The genetic relationships among the analysed genotypes inferred from concatenated, but not edited, sequences were done using the Neighbour-net analysis (NN). Neighbour-net graphs were constructed using SplitsTree ver. 4.10 under uncorrected P-distances and with default settings [81].

Secondly, phylogenetic relationships among the analysed strains were evaluated using maximum likelihood (ML) analysis. Prior to ML, we delimited the position of both intergenic spacers and the 5.8 gene of the ITS region, as well as introns and exons in CaM and benA. Best-fit partitioning schemes for all 15 partitions detected were computed in PartitionFinder ver. 2.1.1 [82] using a greedy search mode. The partitioned dataset (twelve partitions, not shown) was further analysed using ML analysis in RAxML 8.2.12 [83]. The analysis was run with the GTR + Γ + I evolutionary model and 1000 rapid bootstrap inferences. The analysis was computed on the CIPRES Portal ver. 3.3 [84].

### 5.5. Determination of Ochratoxin A (OTA) Level in Vine Fruits

The level of OTA in all 20 samples was estimated using a commercial ELISA kit (AgraQuant Ochratoxin Assay 2/40, Romer labs), following the manufacturer’s recommendations. A mass of 20 g of dried vine fruits per sample was placed in an extraction solution composed of 70 mL methanol and 30 mL water. Filtered extracts were directly used for ELISA analysis with the following settings: detection limit at 1.9 µg/kg, quantification limit at 2 µg/kg and quantification range at 2–40 µg/kg. The intensity of the solution colour in the microtiter wells was measured optically with an ELISA reader, with an absorbance filter of 450 nm. The optical densities (OD) of the samples were compared to those of the kit’s standard.

### 5.6. OTA, FB1 and FB2 Production Ability of Aspergillus Isolates

The ability of the isolates to produce OTA, as well as fumonisins FB1 and FB2, was performed following the protocol of [85] with a minor modification. To test in vitro production of the mentioned secondary metabolites, all strains of *A. carbonarius*, *A. luchuensis*, *A. niger* and *A. welwitschiae* were included. Furthermore, five strains of *A. tubingensis* were also analysed, including both strains from Chilean sample, in which the highest level of OTA contamination was detected, plus three additional randomly chosen strains. Nevertheless, we are aware of the fact that *A. tubingensis* was reported to be a species without ochratoxigenic potential [73]. All tested isolates were grown on Czapek yeast autolysate agar [4] for ten days at 25 °C, and two agar plugs (6 mm diameter) were removed from the centre and the edge of the colony. The agar plugs were weighted, and OTA was extracted according to the QuEChERS method proposed by [86]. The extracts were analysed by ultra-high-performance liquid chromatography (UHPLC Dionex Ultimate 3000 HPLC system, Thermo Fisher Scientific) using a Luna Omega 1.6 µm (50 × 2.1 µm) column [87]. The eluent consisted of H_2_O (phase A) and MeOH (phase B) containing 0.1% formic acid and 5 mM ammonium formate. The gradient elution program for LC prior to Orbitrap HRMS analysis was developed as follows: 0–1 min–0% of phase B, 2 min–95% of phase B, 2.5 min–95% of phase B, 5 min–75% of phase B, 6 min–60% of phase B, 6.5 min–0% of phase B and 1.5 min–0% phase B for equilibrating the column. The flow rate was set at 0.4 mL/min. A total of 5 µL of the sample was injected. Detection was performed using a Q-Exactive mass spectrometer. The mass spectrometer was operated in both positive and negative ion mode using fast polarity switching by setting two scan events (full ion MS and all ion fragmentation (AIF)). Full scan data were acquired at a resolving power of 35,000 FWHM at m/z 200. The ion source parameters were: spray voltage 4 kV (−4 kV in ESI− mode); capillary temperature 290 °C; S-lens RF level 50; sheath gas pressure (N2 > 95%) 35, auxiliary gas (N2 > 95%) 10, and auxiliary gas heater temperature 305 °C. The value for automatic gain control (AGC) target was set at 1 × 106, a scan range of m/z 100 to 1000 was selected and the injection time was set to 200 ms. The scan rate was set at 2 scans/s. For the scan event of AIF, the parameters in the positive and negative ion mode were: mass resolving power = 17,500 FWHM; maximum injection time = 200 ms; scan time = 0.10 s; ACG target = 1 × 105; scan range = 100–1000 m/z, isolation window to 5.0 m/z, and retention time window to 30 s. LOD and LOQ for OTA was 0,06 ppb resp. 0,2 ppb. Details on the OTA, FB1 and FB2 production abilities of the analysed black spore aspergilli strains are summarised in Appendix A.

## Figures and Tables

**Figure 1 toxins-12-00592-f001:**
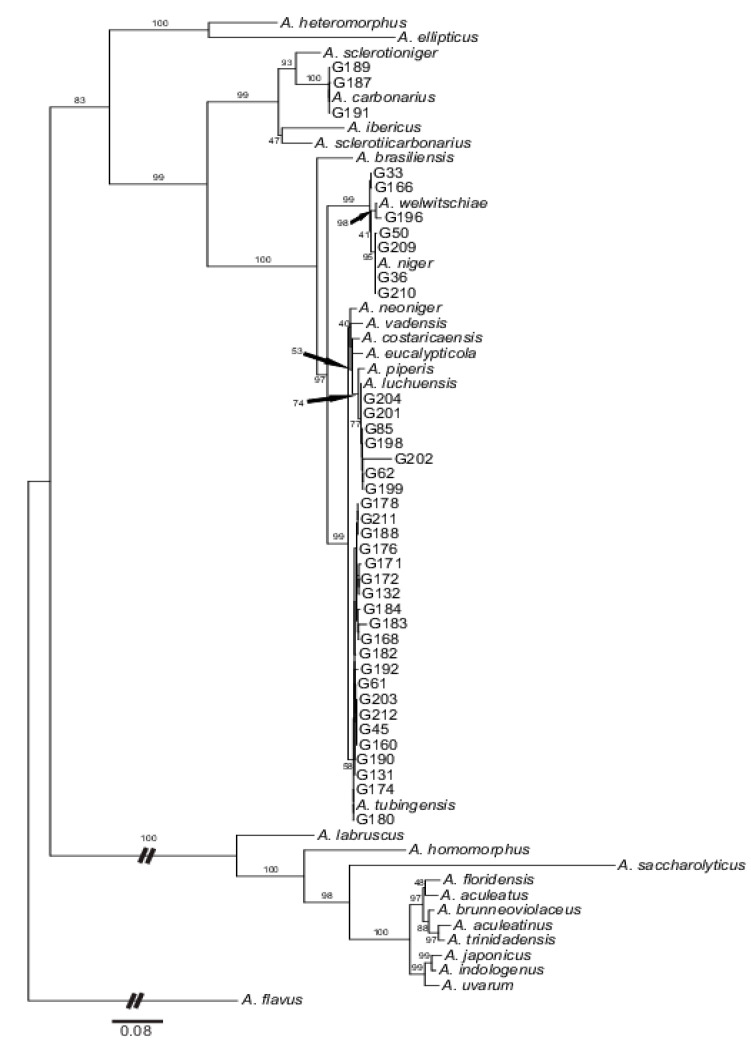
Majority-rule consensus tree as inferred by the maximum likelihood analysis and based on the concatenated dataset including ITS1-5.8s-ITS2, calmodulin and beta-tubulin sequences. Numbers above branches refer to the bootstrap support as inferred for the maximum likelihood analyses (values ≥50% are shown). All species’ accessions from the *Aspergillus* section *Nigri* are represented by the National Centre for Biotechnology Information deposited type strains (Appendix A). For details on accession codes of our strains, see Table 1.

**Table 1 toxins-12-00592-t001:** Details on the origins of the analysed dried vine fruit samples, their taxonomic affiliation and codes of isolated fungal strains and OTA content detected. Abbreviations: RSA = Republic of South Africa; b.d.l. = below the detection limit.

Sample ID	Country of Origin	Level of OTA	Colony Forming Units	Occurrence of *Aspergillus* spp. within CFU	*Aspergillus* Sect. *Nigri*	Strain Code	Another Fungal Taxa Isolated
		(µg/kg)	(CFU)/g				
DVF_01/2016	Chile	b.d.l.	2.2 × 10^3^	84%	*A. niger*	**G_209**	*Penicillium* sp.
					*A. tubingensis*	**G_45**	*Rhizopus* sp.
						**G_178**	
						G_211	
DVF_02/2016	Iran	b.d.l.	5.4 × 10^2^	100%	*A. tubingensis*	**G_180**	-
						G_182	
DVF_03/2016	Iran	1.7	2.2 × 10^3^	88%	*A. niger*	**G_33**	*Aspergillus flavus*
					*A. welwitschiae*	**G_36**	*Rhizopus* sp.
DVF_04/2016	Chile	1.6	9 × 10^2^	80%	*A. luchuensis*	**G_202**	*Rhizopus* sp.
						**G_201**	
						**G_204**	
					*A. tubingensis*	G_203	
DVF_05/2016	Czech Republic	1.6	1.4 × 10^3^	88%	*A. luchuensis*	**G_198**	*Cladosporium cladosporioides*
						**G_199**	*Penicillium* sp.
					*A. welwitschiae*	**G_196**	
DVF_06/2016	Slovak Republic	b.d.l.	9.9 × 10^2^	100%	*A. luchuensis*	**G_62**	-
					*A. niger*	**G_50**	
DVF_07/2016	Turkey	b.d.l.	1.1 × 10^3^	77%	*A. tubingensis*	**G_132**	*Penicillium* sp.
							*Rhizopus* sp.
							*Saccharomyces* sp.
DVF_08/2016	Turkey	1.6	2.7 × 10^2^	0%	*-*	-	*Aspergillus flavus*
							*Cladosporium cladosporioides*
							*Trichothecium roseum*
DVF_09/2016	Chile	10.5	6.3 × 10^2^	14%	*A. tubingensis*	**G_160**	*Alternaria alternata*
							*Paecilomyces* sp.
							*Penicillium chrysogenum*
							*Saccharomyces* sp.
DVF_10/2016	Chile	1.8	1.2 × 10^3^	42%	*A. welwitschiae*	**G_166**	*Rhizopus* sp.
							*Saccharomyces* sp.
DVF_11/2016	unknown	0.8	0	0%	*-*	-	-
DVF_12/2016	Iran	1.8	3.6 × 10^3^	10%	*A. tubingensis*	G_168	-
DVF_13/2016	Chile	10.6	4.5 × 10^2^	40%	*A. tubingensis*	**G_171**	*Penicillium* sp.
						G_172	*Saccharomyces* sp.
DVF_14/2016	RSA	2.5	6.2 × 10^2^	71%	*A. tubingensis*	G_183	*Saccharomyces* sp.
						G_184	
DVF_15/2016	Iran	1.2	1.9 × 10^3^	91%	*A. carbonarius*	**G_187**	*Rhizopus* sp.
					*A. tubingensis*	G_188	
DVF_16/2016	Turkey	1.8	1.9 × 10^3^	91%	*A. carbonarius*	**G_191**	*Rhizopus* sp.
					*A. tubingensis*	G_190	
						G_192	
DVF_17/2016	Czech Republic	b.d.l.	0	0%	*-*	-	-
DVF_18/2016	Uzbekistan	2.1	1.1 × 10^3^	100%	*A. tubingensis*	G_174	-
DVF_19/2016	Chile	3.9	9.9 × 10^2^	100%	*A. niger*	**G_210**	-
					*A. tubingensis*	G_61	
DVF_20/2016	unknown	1.3	4.5 × 1^2^	100%	*A. tubingensis*	G_176	-

Strains in bold were selected for individual analyses for toxigenic ability using HPLC (see Methods).

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
