# Peer review of "Genetic Diversity, Ochratoxin A and Fumonisin Profiles of Strains of *Aspergillus* Section *Nigri* Isolated from Dried Vine Fruits"

_toxins, 2020, doi:10.3390/toxins12090592_

Round 1
Reviewer 1 Report
Overall, this work adds to our understanding of the risk of OTA contamination to these food types, and the most likely fungal source.
This paper does a good job of supporting the stated objectives of the study.
My major comment involves the strict focus on OTA and that analysis was performed on a relatively small number of samples (20).
1) Some of these species, notably A. niger and some strains of A. welwitschia, are known producers of fumonisin, which is also known to be a contaminant. Although the work was designed to solely look for OTA, I think the impact of the work for the larger mycotoxin field would be greatly improved by also including if the tested strains can produce FB.
Based on the LC-MS method described, this could be accomplished using the existing dataset, without performing additional experiments. Fumonisin B2, B4 and B6 would all be easily detected in positive [M+H]+ or negative [M-H]- mode under the described conditions. In negative mode, all major fumonisin produce a distinctive product anion at m/z 157.0143 (see Norman Massbank spectral library). Even if the data is solely included briefly in results and supporting information, I believe it is important to include especially since it would require no additional experiments be performed.
2) I disagree with reporting the presence of OTA in agar plugs as (ug/mass agar). This could easily be altered by the thickness of the agar and therefore the absolute quantity is somewhat meaningless. A simple (+), or (-) designation for presence and absence is preferred here.
3) An explanation regarding the temperatures chosen to culture the strains should be stated. The ideal temperatures for OTA production is likely different for A. carbonarius and A. niger OTA producers.
4) Minor typo errors include:
L24: However, they are often exposed …
L31 “strictly monitored mycotoxins”
L34 “Correct determination of black Aspergillus.
Author Response
Responses to reviewer
Dear editor,
We are deeply thankful to you and both anonymous reviewers for valuable comments that will undoubtedly increase the overall level of our manuscript.
We accepted all criticism and suggestions arisen by reviewers and incorporated changes into the new, revised version of the manuscript. Our responses to reviewer comments are in bold.
Reviewer 1
My major comment involves: the strict focus on OTA.
Some of these species, notably A. niger and some strains of A. welwitschia, are known producers of fumonisin, which is also known to be a contaminant. Although the work was designed to solely look for OTA, I think the impact of the work for the larger mycotoxin field would be greatly improved by also including if the tested strains can produce FB. Based on the LC-MS method described, this could be accomplished using the existing dataset, without performing additional experiments. Fumonisin B2, B4 and B6 would all be easily detected in positive [M+H]+ or negative [M-H]- mode under the described conditions. In negative mode, all major fumonisin produce a distinctive product anion at m/z 157.0143 (see Norman Massbank spectral library). Even if the data is solely included briefly in results and supporting information, I believe it is important to include especially since it would require no additional experiments be performed.
- We agree that the strict focus on OTA might somehow narrow the overall attractivity of our mycotoxicological study. Indeed, our primary intention was to focus on OTA detection because it is the only strictly regulated mycotoxin with specified limits in this kind of food commodity. Nevertheless, and following the recommendation of the reviewer, we inc of tests focused on the ability to produce fumonisins B1 and B2 production detected by our black aspergilli strains. The outcomes of FB’s analyses are now mentioned across the body of the text, and specific results are alos reported in Supplementary data (see Table S3). We agree that the inclusion of this data will strengthen the outcomes of our investigation and make it more attractive for reviewers and potential readers.
that analysis was performed on a relatively small number of samples (20)
- Yes, we agree that the number of samples might have an impact on a study finding and final interpretations. The more robust sampling would be more desirable; however, the number of samples in the study was not restricted by our subjective selection of samples but their availability at the markets. Indeed, used raisins samples represent all raisins products brands available at Slovak food markets at the sampling period of our study.
I disagree with reporting the presence of OTA in agar plugs as (ug/mass agar). This could easily be altered by the thickness of the agar and therefore the absolute quantity is somewhat meaningless. A simple (+), or (-) designation for presence and absence is preferred here.
- We surely understand this reviewer concerns, but we assume that the expression of the presence of OTA in agar plugs as ug (toxin)/kg (agar) is not meaningless or misleading. We strictly followed the manufacturer's protocol recommendations for ultra-high-performance liquid chromatography which specified that final amounts of OTA should be reported in ug/kg. Agar plugs (6 mm diameter) were removed from the centre and the edge of the colony, the plugs were then carefully weighted, and OTA was extracted according to the QuEChERS method proposed by [da Cruz, C.L.; Delgado, J.; Patriarca, A.; Rodríguez, 2019]. Subsequently, the final ratio was computed from the detected OTA amount and weight of the agar plug. The thickness of the agar plug thus should not play any role in the computation of this ratio or in the final interpretation of the data or their comparison with similar studies. The mycotoxins production in vitro cultivated fungi use to be standardly reported in the way of ug/kg (see also Susca et al. 2016 – Frontiers in Microbiology or Pantalides et al. 2017 – Food Microbiology)
An explanation regarding the temperatures chosen to culture the strains should be stated. The ideal temperatures for OTA production is likely different for A. carbonarius and A. niger OTA producers.
- The selection of cultivation temperature in general follows findings of Pitt, Hocking, 2009, who showed that the optimal growth of black aspergili is in the range from 6-47 °C. We decided to select the optimal temperature in the middle of this range, i.e. 25 °C. Indeed, this temperature is standardly used for studies focused on in vitro production of OTA and fumonisins by black aspergili (see e.g. Merlera et al. 2015 – International Journal of Microbiology; Susca et al. 2016 – Frontiers in Microbiology Pantalides et al. 2017 – Food Microbiology)
Minor typo errors include:
L24: However, they are often exposed …
- Corrected following the reviewer recommendation.
L31 “strictly monitored mycotoxins”
- Corrected following the reviewer recommendation.
L34 “Correct determination of black Aspergillus
- Corrected following the reviewer recommendation.

Reviewer 2 Report
The contamination of food products by mycotoxins is a health risk for consumers, so the topic of the publication is very important and still valid.
The literature has been properly selected and the introduction introduces the presented research.
The article requires corrections and supplements. All my remarks are included as comments and underlines in the text.
I miss conclusions in the publication.
In my opinion the analytical methods were not always well selected and the results were presented. It cannot be written that the toxin content was 0 but that it was below the detection limit.

Author Response
Responses to reviewer
Dear editor,
We are deeply thankful to you and both anonymous reviewers for valuable comments that will undoubtedly increase the overall level of our manuscript.
We accepted all criticism and suggestions arisen by reviewers and incorporated changes into the new, revised version of the manuscript. Our responses to reviewer comments are in bold.
Reviewer 2
I miss conclusions in the publication.
- Conclusions were included after discussion section
L19: please inser the „XX“ samples
- The number of the samples was included in to the text
L28: deleted words „spoilers and“
- Both words were deleted from the text
L32: I guess it just means wine. Wine is not considered as food, dried wine fruits means raisins. Please rebuilt this sentence
- The sentence was corrected accordingly.
Table 1. „0“ or below the detection limit?
- The symbol ‚0‘ was replaced the more apropriate abbreviation ‚ b.d.l.‘ ‚below detection limit‘
Table 1. On what basis these strains were selected?
- Yes we agree, the sampling strategy was not clealry explained so we clarified in more detail our sampling strategy for OTA nd FB’s production ability in Material and Methods chapter (p. 11, lns 332 - 338).
L116: In my opinion, the region of origin of the raisins is more important than their production. The contamination of raisins from the Czech Republic and Slovakia is secondary contamination from the production area e. g. air, technological line
- Yes, it is true; our formulation is a bit misleading. The sentence was reformulated as follows:
‚Indeed, the OTA contamination has been found in all world regions from which our raisins samples originated‘.
L121: which strains?
- the specific number and codes of strains were included in the text:
‘Out of all analysed isolates of Aspergillus section Nigri, only two isolated strains of A. carbonarius (G_187, G_191) were found to be ochratoxigenic (Supplementary material Table S3).’
L131: maybe it worth pointing out these are yeasts
- Yes we agree. The text was corrected accordingly:
‘Micromycetes colonising dried vine fruits samples analysed in this study belonged to seven genera of microscopic filamentous fungi, specifically Alternaria, Cladosporium, Aspergillus, Paecilomyces, Penicillium, Rhizopus and Trichothecium. In addition, we also isolated yeasts from the genus Saccharomyces.’
L151: I think the 20 samples tested are not enough to generalize
- Yes, it is true. The more robust sampling would be more desirable; however, the number of samples in the study was not restricted by our subjective selection of samples but their availability at the marked. Indeed, used raisins samples represent all raisins products brands available at Slovak food markets at the sampling period of our study.
To overcome generalisation not supported by robust sampling we change the title of the paragraph as follows: ‘OTA contamination of analysed dried vine fruits samples’ and also included one sentence in to the this chapter itself:
‚ We are, however, a bit reluctant to broadly generalise our findings, whereas, the precise percentage of OTA contaminated samples in our study might be affected by a rather restricted number of analysed raisins samples.‘
L155: Please also refer to RASFF
- We included two sentences reffer in to RASFF : ‘The majority of notifications reported in the RASFF (Rapid Alert System for Food and Feed) concerned aflatoxin B1 (80%), followed by aflatoxins (not specified) (13%) and OTA (5%). The notification trend of OTA during 2007-2016 shows a significant increase, with a peak of notifications in 2016. (Data from Portal of RASFF) [39]’
L177: Was there a correlation between the presence of A. carbonarius in specific samples and OTA contamination of these samples?
- There was no correlation. This information was included into the text.
Driedwine fruits deleted
- Expresion ‚Driedvine fruits‘ were deleted replaced by ‚raisins‘
L216: DG18 medium is better for dried fruit with low water activity
- Yes, this is also true. However, we used malt-extract agar which is commonly used in such kind of studies since it is suitable for a broad spectrum of microscopic fungi Importantly, this kind of a microbiological medium is commonly used in this type mycotoxicological studies.
L284: The reference method for mycotoxin determination is HPLC. Why was this method not used, even though i tis used in the next point?
- ELISA method is a standard method used in food mycology for detection of main mycotoxins. Our primary aim was to detect the most critical mycotoxin (OTA) directly in raisins samples by a routine method. HPLC method was used for a precise definition of multimycotoxin profile (OTA and fumonisins) production in the case of specific isolated strains.
L288: Whether this detection limit was set by the manufacture of the test or by the authors for this food matrix. What is the recovery for this method and this matrix
- This detection limit was done by the manufacture. This commercial ELISA kit (AgraQuant Ochratoxin Assay 2/40, Romer labs) is used for routine detection, especially for detection of ochratoxins. The Ochratoxins ELISA is an immunoassay for the detection of total ochratroxins (A/B/C). This test is suitable for the quantitative and/or qualitative detection of ochratoxins.
L299: Is a pity that the authors did not use the natural food matrix. Conditions in the microbial medium are different than in natural raw materials.
- Yes, it might be exciting to test mycotoxins production on the natural food matrix. However, such experiments were beyond the scope of our investigation, and importantly, the utility of artificial microbial media is standardly used in such kind of studies (see e.g. Merlera et al. 2015 – International Journal of Microbiology; Pantalides et al. 2017 – Food Microbiology)
L303: please insert the appropiate references
- Reference was included into the text.

Round 2
Reviewer 2 Report
All my comments were taken into account and all doubts were cleared. I accept the authors' responses.
I think that the article in its current form could be published on Toxins, but after minor revision and clarification of two points.
(i) In the original version of the article, the authors did not mention the determination of fumonisin in fungal cultures. What is the reason this toxin suddenly shows up now? Why was it not determined in raisins as well? The production of fumonisin is not common among the fungi from Aspergillus genera , it is mainly produced by Fusarium species. This information should be included in the introduction.
(ii) Please explain where the unit (μg/kg) in which toxin content was expressed came from. "Kg" of what? Medium?
Author Response
Responses to reviewers
Dear editors,
we would like to thanks to Reviver 2 for the second round of review and further tuning of our manuscript. We accepted all criticism arisen by reviewer 2 and incorporated changes into the new, revised version of the manuscript or answer his/her questions (see below). Our responses to reviewer comments are in bold. Please note we also done additional minor changes in References to improve their final formatting.
Reviewer 2
All my comments were taken into account and all doubts were cleared. I accept the authors' responses.
I think that the article in its current form could be published on Toxins, but after minor revision and clarification of two points.
- In the original version of the article, the authors did not mention the determination of fumonisin in fungal cultures. What is the reason this toxin suddenly shows up now?
Yes, indeed, the previous version did not contain analyses on FB1 and FB2. The inclusion of those analyses was done as a response to one of the comments of reviewer 1 who requested to include also analyses on FB1 and FB2 into the manuscript. Here is the original statement of reviewer 1 and our response:
Reviewer 1: My major comment involves: the strict focus on OTA. Some of these species, notably A. niger and some strains of A. welwitschia, are known producers of fumonisin, which is also known to be a contaminant. Although the work was designed to solely look for OTA, I think the impact of the work for the larger mycotoxin field would be greatly improved by also including if the tested strains can produce FB. Based on the LC-MS method described, this could be accomplished using the existing dataset, without performing additional experiments. Fumonisin B2, B4, and B6 would all be easily detected in positive [M+H]+ or negative [M-H]- mode under the described conditions. In negative mode, all major fumonisin produce a distinctive product anion at m/z 157.0143 (see Norman Massbank spectral library). Even if the data is solely included briefly in results and supporting information, I believe it is important to include especially since it would require no additional experiments to be performed.
We agree that the strict focus on OTA might somehow narrow the overall attractiveness of our mycotoxicological study. Indeed, our primary intention was to focus on OTA detection because it is the only strictly regulated mycotoxin with specified limits in this kind of food commodity. Nevertheless, and following the recommendation of the reviewer, we include tests focused on the ability of our strains to produce fumonisins B1 and B2. The outcomes of FB’s analyses are now mentioned across the body of the text, and specific results are also reported in Supplementary data (see Table S3). We agree that the inclusion of this data will strengthen the outcomes of our investigation and make it more attractive for reviewers and potential readers.
- Why was it not determined in raisins as well?
Our primary aim was to detect the most critical mycotoxin (OTA) directly in raisins samples by a routine method. Subsequently, we used HPLC method for a precise definition of multimycotoxin profile (OTA and fumonisins) production in the case of specific isolated strains.
- The production of fumonisin is not common among the fungi from Aspergillus genera , it is mainly produced by Fusarium species. This information should be included in the introduction.
Accepted the note about primary production of fumonisins by Fusarium was include in to the introduction part:
The fumonisins are known to be produced predominantly by specific members of the genus Fusarium [e.g. 2,13]. Nevertheless, their production has been evidenced also in some of the Aspergillus species, specifically in A. niger and rarely also in A. welwitschiae to date [17- 20].
- Please explain where the unit (μg/kg) in which toxin content was expressed came from. "Kg" of what? Medium?
In ELISA method the unit μg/kg was expressed for μg /kg for raisins following manufacturer´s recommendations.
In HPLC method, our quantitative evaluation was based on agar dishes. So, the unit μg /kg is related to medium used in other words on agar dish used to growth the fungi.